



# HGS-PDAF (version 1.0): A modular data assimilation framework for an integrated surface and subsurface hydrological model

Qi Tang[1,2], Hugo Delottier[1], Wolfgang Kurtz[3], Lars Nerger[4], Oliver S. Schilling[1,2,5], Philip Brunner[1]

[1]Centre for Hydrogeology and Geothermics (CHYN), University of Neuchâtel, Neuchatel, 2000, Switzerland
[2]Hydrogeology, Department of Environmental Sciences, University of Basel, Basel, 4056, Switzerland
[3]German Meteorological Service, Centre for Agrometeorological Research, Branch Office Weihenstephan, Freising, 85354, Germany
[4]Alfred-Wegener-Institut, Helmholtz Zentrum für Polar- und Meeresforschung, Bremerhaven, 27570, Germany
[5]Eawag, Swiss Federal Institute of Aquatic Science and Technology, Dübendorf, 8600, Switzerland

*Correspondence to*: Qi Tang (qi.tang@unine.ch)

**Abstract.** This article describes a modular ensemble-based data assimilation (DA) system, which is developed for an integrated surface-subsurface hydrological model. The software environment for DA is the Parallel Data Assimilation Framework (PDAF), which provides various assimilation algorithms like the ensemble Kalman filters, nonlinear filters, 3D-
Var, and combinations among them. The integrated surface-subsurface hydrological model is HydroGeoSphere (HGS), a physically based modelling software for the simulation of surface and variably saturated subsurface flow, as well as, heat and mass transport. The coupling and capabilities of the modular DA system are described and demonstrated using an idealized model of a geologically heterogeneous alluvial river-aquifer system with drinking water production via riverbank filtration. To demonstrate its modularity and adaptability, both single- and multivariate assimilation of hydraulic head and soil
moisture observations are demonstrated in combination with individual and joint updating of multiple simulated states (i.e., hydraulic heads and water saturation) and model parameters (i.e., hydraulic conductivity). The new DA system marks an important step towards achieving operational real-time management of coupled surface water-groundwater systems such as riverbank filtration wellfields based on integrated surface-subsurface hydrological models and data assimilation.

## 1 Introduction

Numerical hydrological models are appropriate decision support tools for water resources management, as they can be used to better understand and predict complex hydrological systems that are dynamically evolving as a result of natural and anthropogenic stresses. When it comes to managing shallow groundwater systems, integrated surface-subsurface hydrological models (ISSHMs) (Sebben et al., 2013) are essential as they simulate all the components of the hydrological cycle and their feedback mechanisms within a single framework (Doherty and Moore, 2020; Islam, 2011; Paniconi and Putti,
2015). ISSHMs provide a physically based and hydrologically consistent simulation of water fluxes across the entire hydrological system (Simmons et al., 2020). This makes ISSHMs robust tools for the simulation of water quantity and





quality, and thus for supporting the prospective management of water resources (Belleflamme et al., 2023; Burek et al., 2020; Du et al., 2012; Paudel and Benjankar, 2022; Yang et al., 2021). Furthermore, ISSHMs also allow the potential impacts of climate change and human activity on the natural water system to be studied (Wada et al., 2017). Examples of

such ISSHMs include MIKE-SHE (Refsgaard et al., 1995), InHM (VanderKwaak and Loague, 2001), IHM (Ross et al., 2005), ParFlow (Kollet and Maxwell, 2006), CATHY (Camporese et al., 2010), and HydroGeoSphere (Aquanty, 2020; Brunner and Simmons, 2012).

As with any modelling approach, complex ISSHMs necessitate the minimisation of uncertainty in both model parameters and predictions. This is ideally achieved through inverse estimation of model parameters using available direct and indirect

observations, facilitated by some form of data assimilation. Model parameters in ISSHMs generally represent the many different physically and sometimes also the biogeochemically relevant hydraulic and hydrological properties of the surface and subsurface, and these parameters are typically spatially and sometimes also temporally highly heterogeneous. Moreover, the true values of these parameters are usually not precisely measurable, and any hydrological modelling effort, be it based on an elaborate ISSHM or even a simple bucket-type model, inevitably starts off with considerable prior uncertainty (Moges

et al., 2020). In addition, model forcing data and model structure are associated with uncertainty, and, unless they are reduced and/or appropriately accounted for, all these uncertainties have the potential to significantly limit the reliability of (integrated) numerical models. Thus, quantification and reduction of model uncertainty is a critical step for any decision-based hydrological model (Anderson et al., 2015) and important for both research and for operational modelling efforts (Liu and Gupta, 2007). While different methods exist, one of the most robust approaches to quantify and reduce model

uncertainties is through data assimilation (DA) (Fan et al., 2022). DA is used widely in oceanography and meteorology (see Ghil and Malanotte-Rizzoli, 1991; Hoteit et al., 2018), particularly for global reanalysis (Baatz et al., 2021) and operational weather forecasting (Hu et al., 2023; Navon, 2009), where DA frameworks integrate measurements in near real-time into models and continuously correct for model deviations from the "true" system states. In recent years, DA has also been applied more frequently to continental hydrological systems, especially for experimental studies with physically based

models and operational flood forecasting (Camporese and Girotto, 2022). By continuously incorporating real-time information from ground sensors and remote sensing, as well as weather forecasts, into hydrological models via DA, the uncertainties of hydrological model predictions could be significantly reduced and operational short-term forecasts improved (Di Marco et al., 2021).

So far, studies on the implementation and development of DA for coupled surface-subsurface hydrological systems

modelling, particularly via ISSHMs, are very limited. A successful implementation was demonstrated for the first time by Paniconi et al. (2003), who applied the simple DA method of nudging to the simplified version of the physically based surface-subsurface model CATHY (Camporese et al., 2010). It was shown that through the assimilation of soil moisture observations, the hydrological simulations improved significantly and for little additional computational cost. After more experimental DA examples have been developed with CATHY (e.g., Camporese et al., 2009a; Camporese et al., 2009b), DA

started to be explored also for the use with other ISSHMs. Kurtz et al. (2016) developed a data assimilation framework for



the Terrestrial System Modelling Platform (TerrSysMP) (Shrestha et al., 2014) using the DA software Parallel Data Assimilation Framework (PDAF) (Nerger et al., 2005). TerrSysMP itself is a modular Earth system model consisting of the atmospheric model COSMO (Baldauf et al., 2011), the land surface model CLM (Oleson et al., 2004) and the ISSHM ParFlow (Kollet and Maxwell, 2006), all coupled via OASIS-MCT (Valcke, 2013). The data assimilation framework

TerrSysMP-PDAF allows the assimilation of pressure heads and soil moisture measurements into the ISSHM ParFlow and the land surface model CLM via different assimilation algorithms as provided by PDAF. Similarly, an ensemble Kalman filter (EnKF) based data assimilation system for the physically based ISSHM HydroGeoSphere, EnKF-HGS, was developed by Kurtz et al. (2017), which allowed the assimilation of hydraulic heads with joint updating of both hydraulic heads and hydraulic conductivities. Based on EnKF-HGS, Tang et al. (2017) and Tang et al. (2018) assimilated hydraulic head

observations for the joint estimation of states (hydraulic heads and surface water discharge) and parameters (hydraulic conductivities of an alluvial aquifer and a riverbed). Compared to ParFlow, which is the ISSHM in TerrSysMP-PDAF, and which is best suited for the simulation of larger scale interactions between the subsurface, the land surface and the atmosphere (Condon and Maxwell, 2019), HGS is more suited for local scale surface-subsurface interactions and the explicit and efficient simulation of abstraction schemes in riverbank filtration contexts, reactive transport processes, managed aquifer

recharge systems, geothermal systems, or agricultural drainage (e.g., tile drain) and irrigation infrastructure (Alvarado et al., 2022; Boico et al., 2022; Delottier et al., 2022a; Schilling et al., 2022).

Up to now, only the EnKF was implemented as a data assimilation algorithm for HGS (via EnKF-HGS), and the coupling was neither modular nor user-friendly for, thus not suited for operational implementations. A better solution than coupling a single DA algorithm to an ISSHM is the coupling of an existing DA software that offers a suite of different assimilation

algorithms to choose from and is modular with respect to the choice of states, parameters and observations that should be updated or considered for DA. As a toolbox tailored towards numerical modelling, PDAF offers such a modular choice of widely used DA algorithms and supports both single and multivariate assimilation of different types of observations as well as single or joint state and parameter updating. PDAF also facilitates the addition of novel assimilation algorithms which are not yet included. Owing to its modular design, PDAF makes it very easy to switch between different assimilation methods

without the need for additional coding. Last but not least, the different algorithms are not only fully implemented and optimized but also parallelized, which is a key aspect for the continental-scale hydrological modelling conducted with the TerrSysMP-PDAF platform.

With the aim to provide a DA framework for operational real-time simulations of water quality and quantity in complex systems for which ISSHMs are typically the ideal decision-based modelling tools (e.g., riverbank filtration wellfields,

managed aquifer recharge schemes or agricultural systems), we have developed a highly modular DA framework for ISSHM based on PDAF and HGS. The coupled framework, called HGS-PDAF, is designed to allow updating of integrated flow and transport simulations, and includes the following key features:



1) the most up-to-date and continuously maintained collection of data assimilation algorithms, including the ensemble Kalman filter and its established variants, the ensemble smoother, the 3-D variational method, and the hybrid ensemble-variational method.

2) a modular tool to handle different observation data, in HGS-PDAF currently allowing either single or multivariate assimilation of hydraulic head, soil moisture, and solute concentration measurements.

3) a modular tool to handle different model states and parameters in HGS-PDAF allowing individual or joint updates of one or multiple states (currently: hydraulic heads, soil water saturation and solute concentration) and parameter types (currently: hydraulic conductivity).

4) an open-source code repository, which includes the source code, an example test case and documentation on the use of the code and the execution of the example.

Here, the structure and modules of HGS-PDAF are presented, alongside its capabilities and its performance on a multi-variate, joint state-parameter DA example based on a synthetic alluvial riverbank filtration wellfield model. The structure of this paper is as follows: Section 2 describes the structure of the ISSHM HGS, the DA software PDAF, and the specific DA algorithm used in the illustrative example. Section 3 presents the coupled DA framework HGS-PDAF. Section 4 illustrates the implementation and performance of HGS-PDAF on the synthetic test case. The potential for HGS-PDAF to serve as a DA framework for different scientific and management applications in the water sector as well as avenues for further developments and improvements to HGS-PDAF are discussed in Sect. 5. The source code of HGS-PDAF, a manual as well as the presented example test case are available freely via https://zenodo.org/doi/10.5281/zenodo.10000886 (Tang et al., 2023).

## 2 Hydrological model and data assimilation method

### 2.1 General overview of the ISSHM HydroGeoSphere

HGS (Aquanty, Inc.) is an integrated surface-subsurface hydrological model (ISSHM) that was originally developed by Therrien and Sudicky (1996) and can be used to simulate fully coupled surface water and variably saturated subsurface flow, as well as, heat and mass transport (Aquanty, 2020; Brunner and Simmons, 2012). In HGS, surface water flow is simulated using the two-dimensional Saint-Venant equation and (variably saturated) subsurface flow using the three-dimensional Richards equation. The surface and subsurface domains are fully coupled in a physically consistent manner, enabling dynamic, two-way feedbacks between these two domains. This is achieved by simultaneously solving the surface and subsurface flow and transport equations in one single system of equations. Owing to its versatility, HGS has been used to study surface-subsurface flow and transport in complex, heterogeneous hydro(geo)logical systems (e.g., Ala-aho et al., 2017; Schilling et al., 2014; Schilling et al., 2017; Thornton et al., 2022). It has also been used to assess the potential impacts of, and responses to, climate change on hydrological processes at regional scales (Cochand et al., 2019; Delottier et al., 2022b; Erler et al., 2019; Nagare et al., 2023), to explore the dynamics of coastal groundwater flooding under a dual dual-aquifer





configuration (Tajima et al., 2023), in geophysics to inversely estimate the hydraulic conductivity (Sun et al., 2023) and in the context of supporting hydraulic tomography (Wang and Illman, 2023), and to extract and estimate groundwater recharge (Gong et al., 2023). Importantly, a recent study by Delottier et al. (2022a) has enhanced HGS such that it can now explicitly handle reactive (gas) tracers in transient solute transport simulations under variably saturated conditions.

HGS has three key executables: *grok*, *phgs* and *hsplot*. *grok* is the pre-processing executable which compiles the *prefix.grok*

file containing the model definition and setup information. It prepares all the information needed for HGS to run simulations. *phgs* is the main executable for running a serial or parallel forward numerical simulation with HGS. *hsplot* is the post-processing executable that converts the model output files into a readable format that can be later visualised, for example, by Tecplot (Tecplot, Inc.) or the open-source tool ParaView (Kitware, Inc.). Thus, *grok* must be run before *phgs* is run, and *hsplot* can then be run once the simulations executed by *phgs* have been completed.

Before *grok* is run, as with many numerical models, a number of input files, need to be prepared. These files include a control file, a file containing the model mesh, different parameter definition files, and files containing definitions of boundary and initial conditions. The control file is named *prefix.grok*, where prefix is the user-defined file name. All aspects of the HGS model setup are defined in this file containing the main sections: model grid generation, definition of simulation parameters and material properties, definition of initial and boundary conditions, configuration of (adaptive) time stepping

controls and output controls. The control file also contains the instructions used to build the model files. A detailed description of the available input commands can be found in the HGS reference manual available on the Aquanty, Inc., website ([https://www.aquanty.com/](https://www.aquanty.com/)). When all input files are prepared, *grok* can be executed, which prepares all input files required for the execution of *phgs*. The number of processors to be used during the execution of *phgs* is defined in a default file produced by *grok* and can be manually adapted before executing *phgs*. When running *phgs*, the simulations of the flow

and transport phenomena in the surface and subsurface domains are performed. The output files of *phgs* contain the results for the steady state or transient flow solutions in a set of binary and text-based files. To fully access the simulation outputs, the binary output files must be aggregated and converted by *hsplot* into a composite and readable format.

## 2.2 Data assimilation method and PDAF software

### 2.2.1 Data assimilation and the ensemble Kalman filter

The primary purpose of DA is to sequentially update a model's state by merging it in a statistically optimal manner with the information available from observations or other models, to achieve a physically consistent and optimal representation of the true system state. A state vector in the context of DA refers to the mathematical representation of one or multiple states of a numerical model. In the case of hydrological models, typical states that are considered for updating are hydraulic heads, surface water discharge, soil moisture, evapotranspiration or solute concentrations. In addition to states, model parameters

such as hydraulic conductivity, porosity or soil parameters may also be included in the state vector and thus for updating via





DA. A widely used DA algorithm, the ensemble Kalman filter (EnKF, Evensen, 2003), is briefly described below to illustrate the fundamental procedures of (ensemble) DA.

In ensemble-based data assimilation methods such as the EnKF, the state vector is formulated as an ensemble of the states of multiple different realisations of the same model, each of them representing a plausible state of the system. The state vectors are evolved by running a numerical model forward in time. The resulting spread among the state vectors is used to estimate the probability distribution of the true state of the system. In mathematical terms, consider that a state vector $\mathbf{X}$ can be written as Eq. (1):

$$\mathbf{X}_i = \begin{pmatrix} \mathbf{X}_s \\ \mathbf{X}_p \end{pmatrix}_i \tag{1}$$

where $\mathbf{X}_s$ is the state vector with model state variables and $\mathbf{X}_p$ is the state vector with model parameters. The subscript $i$ refers to the realisation. Considering a forward transient model $M$, the model state at the current time step $t$ can be simulated from the previous time step $t\text{-}1$:

$$\mathbf{X}_{t,i} = M(\mathbf{X}_{t-1,i}) \tag{2}$$

When observations are available at time step $t$, denoted as $\mathbf{y}_t$, they are assimilated. For statistical consistency of the EnKF, the observations are perturbed by a reasonably chosen representative observation error $\boldsymbol{\varepsilon}$. The perturbed observation vector $\mathbf{y}_{t,i}$ is obtained by adding one individual perturbation per realisation $i$ as:

$$\mathbf{y}_{t,i} = \mathbf{y}_t + \boldsymbol{\varepsilon}_{t,i} \tag{3}$$

In the EnKF, the state vector is then updated by combining the observations with the model forecast according to Eq. (4):

$$\mathbf{X}_{t,i}^a = \mathbf{X}_{t,i}^f + \boldsymbol{\alpha}\mathbf{G}(\mathbf{y}_{t,i} - \mathbf{H}\mathbf{X}_{t,i}^f) \tag{4}$$

where $\mathbf{H}$ is the mapping operator matrix (denoted observation operator) between the state vector and the observations, the superscripts $a$ and $f$ refer to analysis (i.e., the updated states) and forecast (i.e., the simulated states), respectively, and $\boldsymbol{\alpha}$ is the damping factor that is used to avoid filter divergence when updating the parameters (Hendricks Franssen and Kinzelbach, 2008), the value varying between 0 and 1. $\mathbf{G}$ is the Kalman gain, which weights the relative importance of the model forecasts and the observations in a Bayesian sense, taking the respective uncertainties into account. The Kalman gain is calculated based on the covariance matrices of the model forecast and the observational error:

$$\mathbf{G} = \mathbf{C}\mathbf{H}^T(\mathbf{H}\mathbf{C}\mathbf{H}^T + \mathbf{R})^{-1} \tag{5}$$

where $\mathbf{C}$ is the covariance matrix of the forecast model states and parameters, and $\mathbf{R}$ is a diagonal covariance matrix that represents the observation errors at individual observation locations. For more details on the EnKF, $\mathbf{R}$, and $\mathbf{C}$, consult Evensen et al. (2022).

### 2.2.2 PDAF features and structures

PDAF (https://zenodo.org/doi/10.5281/zenodo.7861812) (Nerger, 2023) is a software for data assimilation, designed to be used with numerical models. It offers a comprehensive suite of data assimilation algorithms, including ensemble-based





Kalman filters (e.g. the classical EnKF (Burgers et al., 1998; Evensen, 1994), SEIK (Pham et al., 1998), LETKF (Hunt et al., 2007), LESTKF (Nerger et al., 2012)) as well as variational approaches (Bannister, 2017). A comprehensive description of DA methods and variants of the classical EnKF has recently been provided by Vetra-Carvalho et al. (2018). The source code

for PDAF is primarily written in Fortran90, with some features derived from Fortran 2003. Notably, PDAF can be linked to numerical models written in other languages like C, C++, and Python. PDAF's parallelisation features rely on MPI (Gropp et al., 1994) for the software itself, while localised filters additionally support OpenMP parallelisation. Importantly, the core routines are entirely independent of the numerical models, allowing them to be compiled separately and utilized as a library.

To enable a numerical model to perform DA using PDAF, several 'links' must be established between the numerical model

and PDAF. Firstly, in order to effectively combine model simulations and observations, it is necessary to inform PDAF of their relationship in space and time. For example, the observations may not be at the exact location but in the vicinity of where the model grid points are located. Interpolation is required in this case. In addition, it is important to specify how the state vector used in the filter algorithms corresponds to the model variable. For example, to ensure that hydraulic conductivity ($K$) is always positive during the assimilation process, the log-transformed $K$ is considered in the state vector,

but the HGS model uses the $K$ itself. These relationships are outlined in separate subroutines that are provided to the assimilation system by the user. Further details about these subroutines, known as parts of the model bindings, are given in Sect. 3.

When integrating a numerical model with PDAF, there are two different coupling approaches: online and offline coupling. In online coupling, PDAF is integrated directly into a model's source code with the help of the PDAF model bindings.

Conversely, in offline coupling, the PDAF model bindings are compiled independently from the model's source code. Consequently, the numerical model run and the assimilation step are executed as separate processes. The output files generated by the numerical model run serve as inputs to the assimilation step, which produces the updated state vectors (i.e. the analysis $a$) and generates new input files for the next time step to be run by the numerical model.

As an open-source software, PDAF has been coupled with many numerical models. One successful example is its coupling

with the climate model AWI-CM-1.1 (Sidorenko et al., 2015) by Nerger et al. (2020). Using this coupled system, Tang et al. (2020) investigated the role of assimilating oceanic observations on the influence of both the ocean and the atmosphere. This was further extended to carry out strongly coupled DA (Tang et al., 2021) which allows to directly update atmospheric variables through assimilation. PDAF has also been applied to explore DA for the terrestrial system (Kurtz et al., 2016), sea-ice forecasting (Mu et al., 2022; Mu et al., 2020) and climate modelling (Brune et al., 2015).

**3 HGS-PDAF description**

The implementation of HGS-PDAF uses the offline coupling approach. Accordingly, the HGS model has to be restarted after each assimilation step. As this is the case, an otherwise longer transient HGS model must be modified for DA, i.e., reduced to run only the short period between two times with available observations of defined length (e.g., 1 day) at a time, with the





transient forcings split into equally sized intervals that are sequentially applied to this short period model. The length of this
interval is determined by the desired assimilation frequency. The respective procedure is described in detail in Sect. 3.1. The
complete data assimilation workflow as applied to HGS through PDAF is described in Sect. 3.2. In HGS-PDAF, HGS,
PDAF and model bindings are compiled as separate libraries and stored in separate folders, with detailed descriptions of the
respective libraries provided in Sect. 3.3.

### 3.1 Adaptation of the HGS forward model runs for the assimilation run

DA sequentially updates the model states (and if desired model parameters) during the transient model run, with the transient
model being 'interrupted' for updating by DA at specified intervals. This means that for each new forecast step, the model
must be restarted with the parameter fields and state variables that have been updated by the DA as the new initial conditions
for the next time step. Modifications to the HGS model configuration are therefore required. The sequential model is thus
split into a short period model with the transient boundary conditions applied sequentially to this short period model. The
numerical model here always uses the same mesh and the same model structure, but the boundary conditions, parameter files
and initial conditions are replaced after each of the intervals.

### 3.2 Workflow of HGS-PDAF

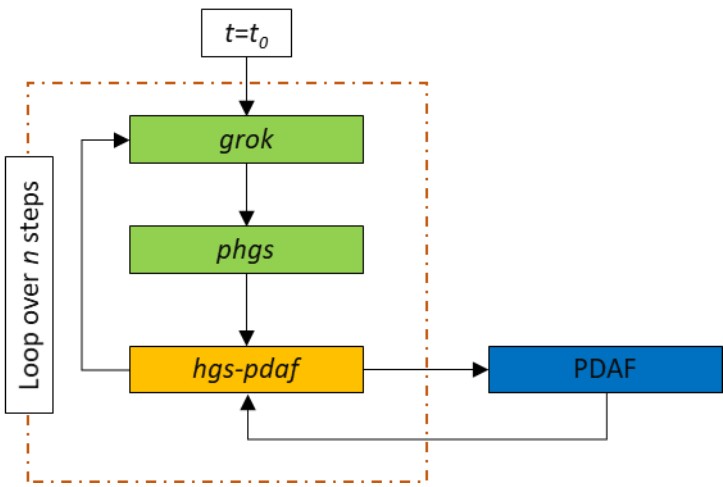

**Figure 1: Flowchart of the overall HGS-PDAF workflow. The green blocks are the parts associated with the HGS model, the**
**yellow blocks are the model bindings that couple HGS to PDAF, and the blue block is the PDAF software itself.**

The overall workflow of HGS-PDAF is illustrated in Fig. 1. First of all, to run HGS-PDAF, a shell execution script called
the 'driver', which is currently implemented for Linux, needs to be prepared. This driver manages the loop in which the HGS
and data assimilation executables are called sequentially throughout the entire run period. At each time step, the driver first
calls the two HGS executables *grok* and *phgs*. After that, *hgs-pdaf*, which is the executable containing the model bindings





that make the connection between HGS and the PDAF (see Sect. 3.3 for details), is called. *hgs-pdaf* checks if observations are available for the current time step *t* and, if there are, calls PDAF to perform DA according to the chosen DA algorithm. As *hgs-pdaf* reads model outputs directly from the *hgs* binary output files, there is no need to call *hsplot*. After computing the DA analysis update, *hgs-pdaf* writes the updated state vector (containing only states or both states and parameters) as new HGS input files for the next time step. In the following, a generic run is described in detail.

Consider a DA run with HGS-PDAF for an ensemble of *m* state realisations and a transient model with a total runtime that splits into *n* timesteps of equal interval $t_{int}$. Before starting the run, an initial ensemble of *m* different model realisations needs to be created. The initial ensemble should account for the uncertainty inherent to the natural hydrological system to be simulated. These realisations can be generated in a number of different ways and take into account several different sources of uncertainty, for example, uncertainty in initial conditions, model parameters, boundary conditions and external forcings.

Subsequently, the run can be started. At the first time step $t=t_0$:

1. The ensemble of HGS models is initialized/pre-processed in parallel by *grok*. The model mesh, boundary conditions, parameters, and initial conditions are checked and read.

2. The ensemble of HGS models is run in parallel for $t_{int}$ by *phgs*. This is the most computationally demanding step, as it requires running all HGS model realisations forward in time. In the current version of *hgs-pdaf*, no model run 260 failure management option is implemented, which requires that all model realisations need to successfully complete for continuation of the run.

3. Now *hgs-pdaf* is executed. Fig. 2 shows the call sequence within *hgs-pdaf*. The steps are as follows:

 3.1 The parallelisation for PDAF is initialized. The MPI commands are defined for the filter, respectively.

 3.2 The data assimilation is initialized. The parameter values for PDAF in the configuration files in the Fortran 265 '*namelist*' format are read. See Sect. 3.3.2 for a detailed description of these configuration files. Next, the dimension of the state vector is determined. The state variables and parameters from step (2) are read from the output files of HGS. Their values, called 'forecast', are entered into the state vector. This is done for each ensemble state.

 3.3 The ensemble mean and standard deviation of the ensemble of state vectors are written to a netCDF file 270 *Output.nc*. In addition, if required, the results for each realisation are written to *m* netCDF files *Output_ens_i.nc*, where *i* represents the realisation.

 3.4 The observations are mapped into the state space by the observation operator. PDAF then performs the analysis step of the data assimilation by integrating the observations with the model forecast according to the chosen DA algorithm, e.g., using the EnKF. The ensemble of state vectors is then updated now holding 275 the 'analysis'.

 3.5 The 'analysis' state information is written into the file *Output.nc* analogous to Step 3.3. The ensemble of 'analysis' state vectors is written into HGS format in parallel for each ensemble member. These files will be used as the initial condition for computing the next time step with HGS.





3.6 PDAF is finalized which completes the execution of *hgs-pdaf* for this analysis time step.

Steps (1) - (3) are repeated until $t = t_{int} \cdot n$. In the current implementation, DA results at every analysis time step are stored in netCDF format while the original output files of the HGS model at the final step are also retained.

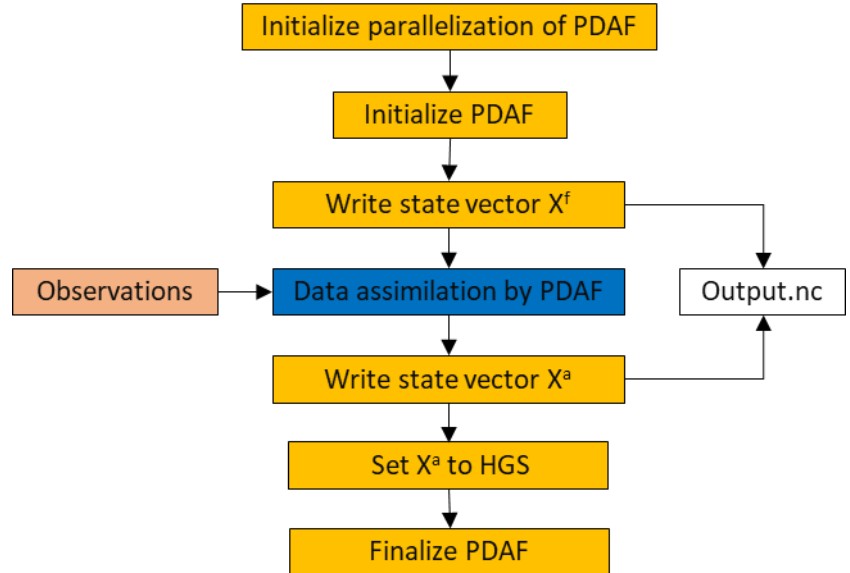

**Figure 2: Call sequence of different subroutines within *hgs-pdaf*.**

### 3.3 Model bindings: *hgs-pdaf*

To couple HGS with PDAF, a number of routines - known as model bindings - provide PDAF with the information from HGS and subsequently pass information from PDAF back to HGS after DA. Since these model bindings are written in Fortran, the following text uses terminology as it is called in Fortran. The *main* program is responsible for calling various HGS and DA subroutines sequentially. The subroutines/modules developed are grouped and described in as follows.

#### 3.3.1 Data assimilation and the ensemble Kalman filter

These subroutines are designed to initialize the parallelisation, parameterisation, and state vector for DA. The MPI execution environment is initialized in *init_parallel_pdaf* at the very beginning. Initialisation of PDAF is done by *init_pdaf*. This includes the following parts as shown in Fig. 3:

(1) Parameters such as the filter type, localisation and inflation are predefined in *init_pdaf*. Parameters specified in *namelist* files are read by *read_config_pdaf*;

(2) The information about the model mesh, such as the total number of nodes and elements in the model, is read in by the *HGS_init* function;

(3) The setup and dimension of the state vector is defined in *initialize*. It is calculated by the details given in (1) and (2). For example, if the state vector, as per the definition in (1), contains the hydraulic heads (i.e., a hydrological system





state, defined for each model node) and *K* (i.e., a hydraulic parameter, defined for each model element), then the

number of nodes for the hydraulic heads is $n_{nodes}$, and the number of elements for *K* is $n_{elements}$, which is defined in

(2). In this case the dimension of the state vector as calculated by *initialize* would be $n_{state} = n_{nodes} + n_{elements}$;

(4) Information about the configuration of the DA, as defined by the initialisation subroutines, can be printed out by

*init_pdaf_info*;

(5) The values for the elements that have been set to define the state vector are also read at this point from the ensemble

of HGS runs that were run forward in time, i.e., the 'forecast'. This is done directly in *init_pdaf*. Variables included

in the state vector can be hydraulic heads, soil water saturation, and solute concentrations (modelled system states)

and *K* (model parameter);

(6) Initialize the DA output netCDF files *Output.nc* by *init_output_pdaf*.

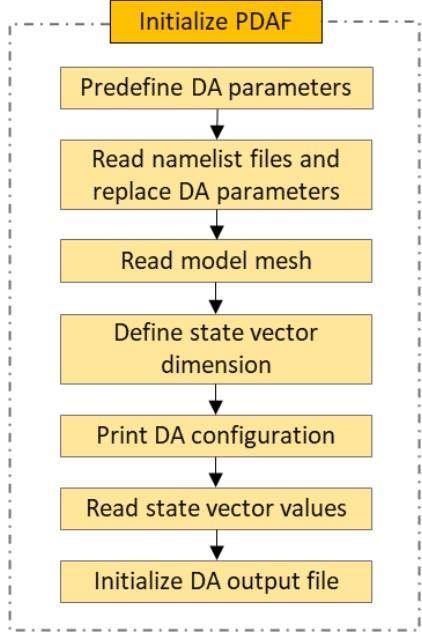

**Figure 3: Flowchart of the initialisation of data assimilation.**

### 3.3.2 Parameterisation modules

The parameters for HGS and DA are predefined in the initialisation phase. However, for each DA application, users should

define them according to their system knowledge and needs. These parameter values are defined in the two *namelist* files,

*namelist.pdaf* and *namelist.hgs,* that are provided by the user. The available parameters that can be defined in the *namelist*

files are described in the Appendix. These two *namelist* files are read by the subroutine *read_config_pdaf* in the initialisation

phase. The parameters used in DA and HGS will then be replaced with the values specified in these *namelist* files instead of

the default values defined in *init_pdaf* by *read_config_pdaf*.



### 3.3.3 Observation modules

For each observation type, a different observation module *obs_VAR_pdafomi* exists. Here, *VAR* refers to the name of each

type of observation, for now hydraulic heads (HEAD), soil moisture (SAT), solute concentrations (CONC), but more can be added swiftly. These different observation modules are independent of each other, allowing for several different types of observations to be modularly combined and either assimilated separately or in a chosen combination. Below, the functioning of the observation modules is described with the example of hydraulic heads.

Like every observation module, the *obs_HEAD_pdafomi* module contains subroutines that initialize the information about

the observations (*init_dim_obs_HEAD*) and to apply the observation operator (*obs_op_HEAD*). Hydraulic head observations are read from the observation input file by *init_dim_obs_HEAD*. The number of observations at the current time step is then counted, which define the dimension of the observation vector for the observation type, in this case hydraulic head. The observations are checked by excluding the unreasonable values (e.g. by defining a threshold value) and the indices of the observations deemed usable during the current time step are stored. The coordinates, the values, and the errors of the

respective observations are also stored.

*Obs_op_HEAD* is the implementation of the observation operator. Thus, it is responsible for the mapping between the state and the observation domains. Various observation operators from PDAF can be selected here by calling the corresponding subroutine *PDAFomi_obs_op_X*. It is also possible for the user to add his own observation operator here. Figure 4 gives an overview of how the observation module works.

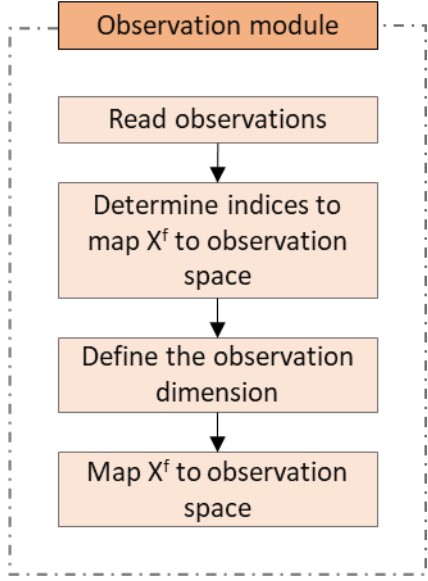


**Figure 4: Illustration of the observation module.**




The subroutine *init_dim_obs_pdafomi* is used to combine the different observations. This subroutine provides an interface between PDAF and the different observation modules. *init_dim_obs_pdafomi* calculates the full dimension of the observation vector by combining all the chosen observation types.

**3.3.4 Assimilation subroutine**

The subroutine *assimilation_pdaf* handles the DA analysis step. The DA algorithm is called according to the filter type defined in the *namelist.pdaf* file. The corresponding filter then updates the variables stored in the state vectors.

**3.3.5 Pre- and post-processing subroutines**

At each time step, the ensemble mean and standard deviations of the state vector at the prediction and analysis stages are
computed by the *prepoststep_ens_offline* subroutine. By default, all these values are written to the output netCDF file *Output.nc*. This is done by the subroutine *write_netcdf_pdaf* in the *output_netcdf* module which contains various subroutines to write the results of the DA into files. In addition, if the user requires the output of all the individual ensemble members, the subroutine *write_netcdf_pdaf_ens* can write the results of each ensemble member to separate netCDF files *Output_ens_i.nc*. Note that these DA output files are different from the output files of the HGS model, which are stored
separately in the original format.

When data assimilation is complete for a time step, an important step is to write the updated states (and parameters) back to the original HGS model format, so that they can subsequently be used as new initial conditions and parameters for the next simulation time step. This is done by calling output subroutines to write files that are compatible with HGS at the end of the main HGS-PDAF program. The development of these output subroutines is beyond the scope of this paper.

**3.4 Scalability of HGS-PDAF**

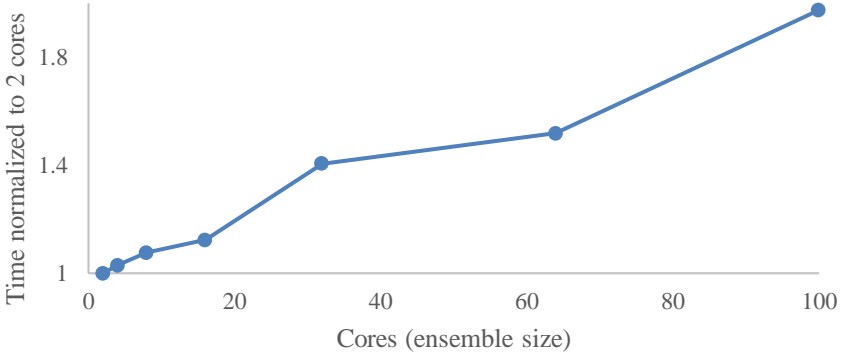

**Figure 5: Computing time of HGS-PDAF on JURECA-DC for different ensemble sizes between 2 and 100 normalised by the time for ensemble size 2.**





The HGS-PDAF code is parallelized with MPI and uses only CPUs. For the scalability test, we used the illustrative example
described in Sect. 4. Different ensemble sizes varying from 2 to 100 members/realisations were tested. Each model was run
on one individual core, thus peaking at 100 cores for the case of 100 ensemble members. All the test runs were carried out on
JURECA-DC CPUs from Jülich Supercomputing Centre in Germany. Figure 5 shows the scaling behaviour of HGS-PDAF
on the JURECA-DC CPUs. The execution time is normalized with the time for an ensemble size of 2 members. When the
ensemble is increased from 2 to 100, the execution time increases by about 50%.

## 4 Illustrative examples of the capabilities of HGS-PDAF

Here, the capabilities of HGS-PDAF are illustrated using a quasi-hypothetical numerical river-aquifer model designed after a
real-world riverbank filtration site in the Swiss pre-Alps which has already served for several studies as a model system for
tracer and DA methods development (see Popp et al., 2021; Schilling et al., 2022; Tang et al., 2018). The model was thus
designed to be representative of an alluvial river-aquifer system where groundwater is pumped for drinking water supply
from wells located in the direct vicinity of a river, inducing so called bank filtration. Such systems are highly suitable for
drinking water production owing to the high $K$ and natural filtration capacity of the alluvial sand and gravel materials which
make up the riverbed and the aquifer. However, in such systems, the interactions between rivers and the underlying aquifers
can be highly dynamic, changing from losing to gaining conditions, and back, within just a few tens of meters, and the
heterogeneity of the alluvial sand and gravel material can be very complex, with irregular paleochannels potentially leading
to strong preferential flow. Without suitable observations and integrated numerical flow and transport models, understanding
and managing such systems becomes a major challenge. Therefore, DA and integrated surface-subsurface hydrological
modelling tools, in particular our HGS-PDAF, are of high interest to continuously update and correct model predictions for
optimal decision support. This quasi-hypothetical model was chosen to demonstrate the capability of HGS-PDAF to
consistently reproduce both system states and parameters even in a highly dynamic and complex hydrological system.

### 4.1 Basic model setup

The real-world alluvial sand and gravel aquifer system, according to which the illustrative model was designed, is
characterized by a distinct paleochannel of well-sorted gravel that exhibits substantially higher hydraulic conductivities
compared to the surrounding, unsorted alluvial sand and gravel sediments (Schilling et al., 2022). The slightly abstracted,
generalized synthetic version of the real-world site model has been introduced by Delottier et al. (2022a) for the
development of environmental gas tracer transport simulations with an ISSHM and efficient data space inversion techniques
for complex heterogeneous aquifer systems (Delottier et al., 2023).

Geometrically, the model represents a 3-D rectangular domain with dimensions 500m x 300m x 30m (Fig. 6). A river of 20m
width and 2m depth is explicitly represented in the model at X = 0 m. The horizontal resolution of the finite elements mesh
varies between 4m along the river and riverbanks, and 7m on the alluvial plain. Vertically, the model consists of 14 layers,





with thicknesses ranging from 0.5 to 4m on the alluvial plain, and slightly smaller thicknesses underneath the river and riverbanks. In total, the model consists of 112,240 nodes and 204,000 elements. Two riverbank filtration wells, spaced at 100m and located at a distance of approximately 90m parallel to the river, extract groundwater from a depth of 14m. The heterogeneity in $K$ as typically found in such alluvial river-aquifer systems is implemented via a highly conductive paleochannel.

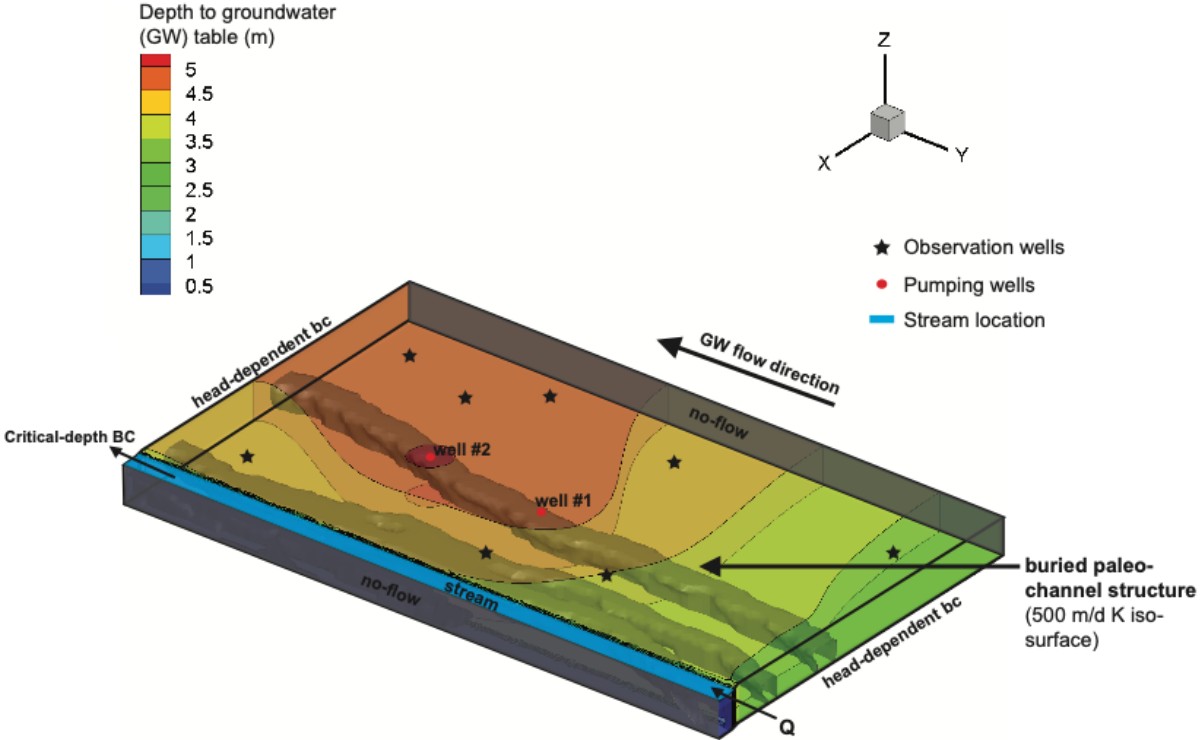


**Figure 6: 3-D view of the model domain and model boundary conditions. Contours represent the groundwater table depth below the surface. Locations of eight virtual observation wells are marked as stars. The location of the highly conductive paleochannel is indicated by a 500 m/d iso-surface for $K$.**

In the surface domain, constant boundaries are set for the upstream with an inflow rate of 1.71 m³/s. A critical depth

boundary is set as a boundary condition for the outflow of the stream. In the subsurface domain, a head-dependent Cauchy-type boundary condition was applied to the groundwater flow. At the upstream, a constant hydraulic head of 99.5m was assumed, while at the downstream, a constant hydraulic head of 93.2m was considered. The conductance for these boundaries was set to 5.8 m²/s. The model was forced with transient boundary conditions to reproduce a controlled pumping experiment in which pumps are first running at a constant rate of 400 m³/h for 15 days and are subsequently turned off for 50

days, after which they are again turned back on to a constant rate of 400 m³/h for the remaining 30 days of the experiment (Fig. 7). A coupling length of 0.001 m is used to account for the exchange fluxes between the surface and the subsurface





domains. Figure 6 shows a 3-D view of the model domain, boundary conditions and paleochannel location, while Fig. 7 shows a schematic of the transient pumping rates employed for the experiment.

**Transient pumping**

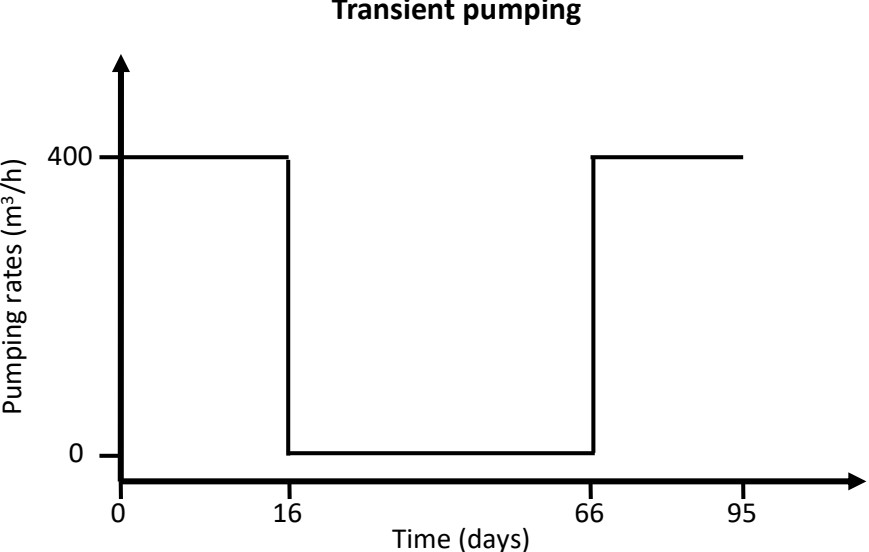

**Figure 7: Transient pumping rates during the simulation period.**

In HGS, time steps are adaptive so that no specific restrictions were applied to maximum time step sizes, with the limitation that the maximum time step size could not be larger than the assimilation time step tint as defined for PDAF. The initial conditions were obtained for each model realisation individually via a 1-year spin up run with constant boundary conditions corresponding to the conditions at the beginning of the 95-days pumping experiment (i.e., with maximum pumping regime).

**4.2 Prior ensemble and synthetic observations**

In this study, the prior uncertainty of the system is characterized by the observed variance of the initially generated ensemble of hydraulic properties, i.e., the $K$. A comprehensive description of the generation of the ensemble is described in Delottier et al. (2023). Briefly, the prior ensemble was developed by using a stochastic alluvial feature generator ALLUVSIM (Pyrcz et al., 2009), geared towards the generation of an alluvial sand and gravel aquifer with distinct paleochannel features. To represent the well-sorted paleochannel and the unsorted surrounding sediments, two categorical parameter fields were created and each of these two categories was populated with a spatially uniform $K$ (i.e., producing two types of sediments with homogeneous properties each). In this way, hydraulic conductivities were parameterized on a model element basis, producing what is known as heterogeneous fields. During DA, the $K$ value of each of these numerical model elements was adjusted. In addition to these heterogeneous $K$- fields, an ensemble of 100 realisations with different homogeneous $K$ values was also considered for the experiments for comparison purpose. These 100 homogeneous $K$ values were defined as the arithmetic averages of the 100 heterogeneous $K$ fields.



To generate synthetic observations against which the performance of DA could be evaluated, one of the stochastic realisations of the ALLUVSIM simulations was defined as a reference heterogeneous *K*-field or synthetic 'truth'. This reference heterogeneous *K*-field is illustrated in Fig. 6. To generate observations for the assimilation experiment, 8 locations
within the model domain were chosen and daily time series of hydraulic heads and soil water saturation at a depth of 1.5m were extracted from this reference simulation. These observation time series were subsequently stochastically perturbed by a normally distributed error with a standard deviation of 5cm for hydraulic heads and 1% for soil water saturation.

### 4.3 Data assimilation scenarios

**Table 1: Overview of illustrative DA scenarios. The open loop scenarios, that is, the scenarios without updating, are labelled 'ol'.**
**For all scenarios with updating, the following naming convention applies: The first part of the name identifies the variables that were included in the state vector (i.e., the variables that were updated), while the second part identifies the observations that were assimilated. h, k and s stand for hydraulic head, *K*, and soil water saturation respectively. Individual damping factors and the conceptualisation of the prior *K* values are also indicated for each scenario.**

| Simulation scenario | Damping factor | Prior *K* values |
|---|---|---|
| ol | - | |
| h_h | 1 | |
| hk_h | 1 | |
| hs_h | 1 | |
| hs_s | 1 | homogeneous |
| hs_hs | 1 | |
| hsk_h | 1 | |
| hsk_s | 1 | |
| hsk_hs | 1 | |
| ol | - | |
| h_h | 1 | |
| hk_h | 0.1 | |
| hs_h | 1 | |
| hs_s | 1 | heterogeneous |
| hs_hs | 1 | |
| hsk_h | 0.02 | |
| hsk_s | 0.02 | |
| hsk_hs | 0.02 | |

To demonstrate the modularity and capability of HGS-PDAF, 20 different DA scenarios that cover combinations between:



- single- and multivariate assimilation of hydraulic heads and/or soil moisture observations,

- updating of either one or a combination of states (i.e., hydraulic heads and/or soil water saturation),

- joint update of a one or a combination of states alongside the parameter *K*, and

- one of two scenarios of prior uncertainty in *K* (i.e., heterogeneous or homogeneous properties),

were run. As a DA algorithm, the EnKF was chosen. In addition, runs without DA (so called 'open loop' runs) were carried out for both the heterogeneous as well as the fully homogeneous *K* scenarios.

Owing to the relatively small size of the simulated system, no localisation was applied. For the heterogeneous *K* scenarios, when *K* was updated with DA, a damping factor of less than 1 was applied. For the homogeneous *K* scenarios, as the assumption of homogeneity already acted as a regularisation for the parameterisation of *K*, the enforced homogeneity in *K*

during the update always produced in a large enough ensemble spread, i.e. the damping factor is equal to 1. Table 1 gives the values of the damping factor used for all DA scenarios.

For all the scenarios, *hgs-pdaf* was run on a highly parallelized Linux cluster so that all individual ensembles in the priors were executed in parallel. It took approximately 11 hours for *hgs-pdaf* to complete one single scenario.

**4.4 Results and discussion of the illustrative DA experiment**

The performance of DA with HGS-PDAF is evaluated by comparing the simulated hydraulic heads and soil water saturation to the synthetically observed hydraulic heads and soil water saturation, respectively. The average relative difference (over the 8 observation wells) between the simulated (represented by the ensemble mean) and synthetically observed states are illustrated in Fig. 8. Results are presented for all the scenarios and for the two different prior ensembles (i.e., homogeneous and heterogeneous *K* fields).

It is remarkably clear from Fig. 8 how DA applied to an ISSHM of a riverbank filtration site (by using HGS-PDAF) is able to reduce the misfit for almost all scenarios and for the two prior ensembles. Overall, the model performance (with respect to both employed observation types) is significantly better when starting from and allowing heterogeneous *K*-fields to arise, compared to when employing the assumption of homogeneity.

For hydraulic heads, the best performance in reducing the model misfit was obtained when assimilating hydraulic heads and

updating both hydraulic heads and *K* (*DA_hk_h*). Little to no improvements were gained by assimilating also soil water saturation alongside hydraulic heads. In the heterogeneous case, scenario *DA_hk_h* performed so well that the averaged ensemble mean model error was reduced to reflect the measurement error (5cm). On the other hand, scenario *DA_hsk_hs*, in which hydraulic heads, soil water saturation and *K* were all updated together based on observations of hydraulic heads and soil water saturation, performed the worst. Even when a low damping factor was used, *K* values did not improve and turned

out highly unrealistic (results not shown). On the other hand, spurious updates of *K* were not observed in the *DA_hsk_hs* scenario when run with the homogeneity assumption, as the homogeneity assumption helps to regularize the problem and ensures consistent estimates of homogeneous *K* fields. The poor performance for updating *K* in the heterogeneous scenario can likely be attributed to the fact that updating categorical prior parameter fields violates the multiGaussian assumption



inherent to the ensemble Kalman filter (Evensen et al. (2022)). In such cases, other methods such as Data Space Inversion
(DSI) or multiple point geostatistics (MPS) should produce better results (Delottier et al., 2023; Remy et al., 2009).

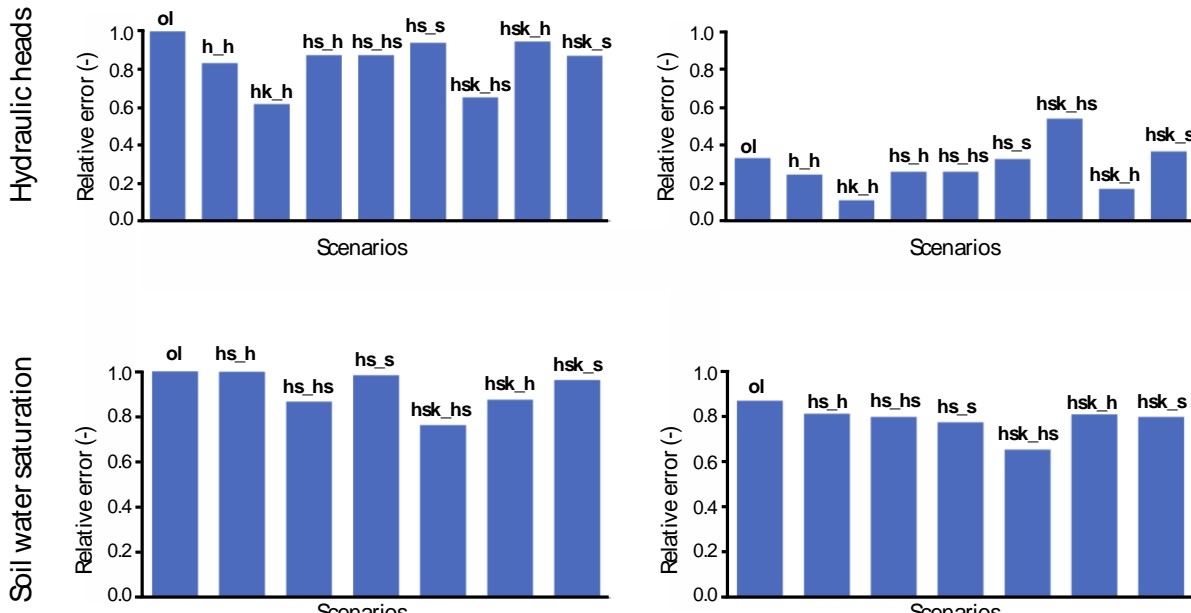

**Figure 8: Ensemble mean of the relative differences (averaged for all observation wells) between simulated and observed states (hydraulic heads and soil water saturation) for (left) the homogeneous scenarios and (right) the heterogeneous scenarios. For the soil water saturation, only scenarios where soil water saturation was updated are shown.**

For soil water saturation, the overall model performance was less improved by DA compared to the improvement for the
reproduction of hydraulic heads. The largest improvement was achieved when hydraulic heads, saturation and $K$ were
updated jointly using both observations of hydraulic heads and soil water saturation (scenario *DA_hsk_hs*). On the other
hand, very little improvement on model performance could be achieved when only observations of soil water saturation were
assimilated, irrespective of the combination of states (and parameters) chosen to be updated. This poor performance of using
soil water saturation observations for DA of an ISSHM is likely explained by the fact that soil water saturation observations
stem from locations relatively far away from the stream and which therefore did not show a strong variation throughout the
pumping experiment. In this specific configuration, the information contained in observations of saturation was thus limited
and could not match up against the information contained in hydraulic heads, which varied strongly throughout the pumping
experiment.

Concerning reproducing the true $K$ field, as long as the $K$ fields were updated from heterogeneous priors and heterogeneous
structures were allowed to arise during updating, a reasonably good overall agreement could already be achieved by only
using hydraulic heads (Fig. 8 and Fig. 9). This is certainly partly owed to the fact that the initially chosen heterogeneous
prior was *a priori* a good approximation of the synthetic truth, as can be directly seen in the two examples illustrated in Fig.
9 as well as in the relatively good performance of the heterogeneous open loop run. As such, this illustrative case highlights





the importance of choosing as good a prior as possible in such heterogeneous $K$ systems, particularly because paleochannel facies are, as outlined previously, difficult to identify from hydraulic head observations alone. Nevertheless, even though the performance of DA was generally good even for $K$, the non-multiGaussian connectivity of the structures could not be preserved perfectly during DA with HGS-PDAF, as can be seen in Fig. 9. As outlined previously, however, this is an expected outcome of DA with a multiGaussian method such as EnKF.

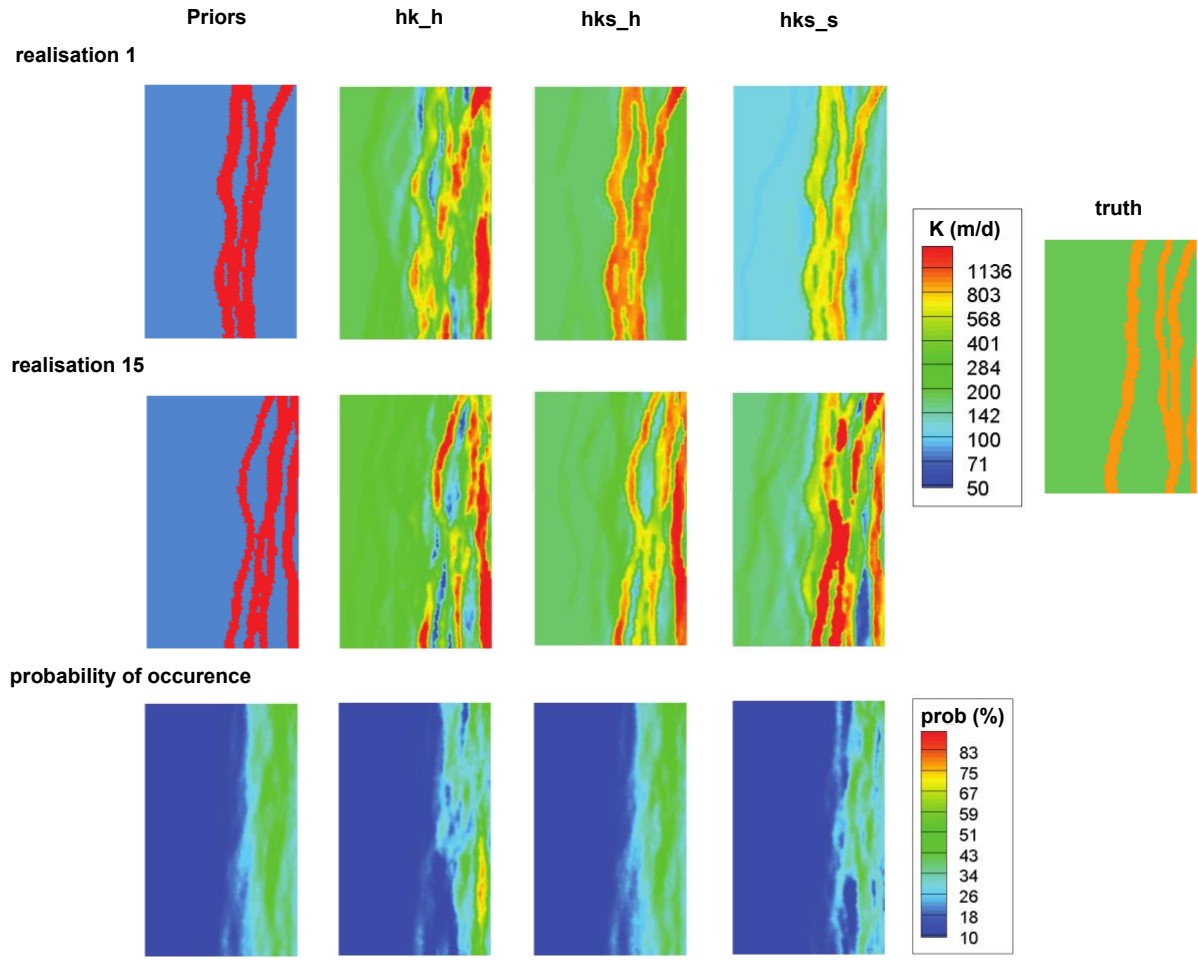


**Figure 9: Posterior estimates of $K$ fields with heterogeneous priors for three different scenarios and two individual realisations. The bottom row indicates the probability of occurrence of a paleochannel, calculated by considering a given threshold (i.e., 600 m/d) above which a buried paleochannel facies is potentially identified.**

## 5 Conclusions

We have here introduced a new data assimilation framework for fully integrated surface-subsurface hydrological models by providing a coupling between the ISSHM HGS and the DA software PDAF. This highly modular DA framework allows for the single and multivariate assimilation of several types of observational data, including hydraulic heads, soil water





saturations and solute concentrations, and individual or joint update of several model states and parameters, including hydraulic heads, soil water saturation, solute concentrations, and hydraulic conductivities. The scalability of HGS-PDAF was

evaluated on the Jülich Supercomputing Centre in Germany and the usability and modularity of HGS-PDAF was illustrated with a synthetic river-aquifer and bank filtration model and the standard ensemble Kalman filter method (one of several DA algorithms provided by PDAF).

Compared to existing hydrological data assimilation systems, the advantage of the newly developed HGS-PDAF lies in its consideration of ISSHM, its large selection of different assimilation algorithms as provided by PDAF, its modularity with

respect to combining observations, states, and parameters to be considered for DA, and the flexibility and ease at which new observations, states and parameters may be added to the already implemented ones. While in the current version of HGS-PDAF only global filters are implemented, the implementation of localized filters is planned for the next iteration.

**Appendix:** *namelist* **files**

The parameters that need to be defined in the *namelist.hgs* file are listed in Table A1. These parameters are used for the HGS

model.

**Table A1: Parameters defined in *namelist.hgs*.**

| Parameter name | Description |
| --- | --- |
| prefix | String, file name of the HGS model |
| insuffix | String, the suffix of the files storing the initial conditions |
| outsuffix | String, the suffix of the output files |
| isolf | 'True' if overland flow is also simulated, and 'false' if only groundwater flow. |
| isconc | 'True' if mass transport is simulated, and 'false' if not. |
| hgs_version | '1' for HGS versions before 2013, and '2' for HGS 2013 and newer versions |

The parameters that can be defined in the *namelist.pdaf* file are listed in Table A2. These parameters are used for data assimilation. If the values of these parameters are not specified in *namelist.pdaf*, default values are used. A detailed description of these parameters can be found from the PDAF website (http://pdaf.awi.de).

**Table A2: Parameters defined in *namelist.pdaf*.**

| Parameter name | Description |
| --- | --- |
| n_modeltasks | Number of parallel model tasks, default is 1 |
| dim_ens | Ensemble size |
| dim_lag | Number of time instances for smoother |
| type_forget | Type of forgetting factor. '0' for fixed, '1' global adaptive, and '2' for local adaptive for LSEIK/LETKF/LESTKF |



| | |
|---|---|
| forget | Values of Forgetting factor |
| type_trans | Type of ensemble transformation. Values differ for local filters. Detail information on these values can be found in the code. |
| type_sqrt | Type of transform matrix square-root. '0' for symmetric square root, '1' for Cholesky decomposition |
| incremental | '1' if incremental updating is performed. Only used in SEIK/LSEIK. |
| step_null | Initial time step of assimilation |
| write_da | Whether to write the output file for DA[*] |
| write_ens | Whether to write the output files for each realisation[*] |
| str_daspec | String to identify assimilation experiment |
| printconfig | Whether to print information on all configuration parameters[*] |
| istep | Real time step for HGS and PDAF |
| screen | Write screen output. '1' for output, and '2' for adding timing information |
| assim_o_head | Whether to assimilation the hydraulic head observations[*] |
| path_obs_head | Path to the file storing the head observations |
| file_head_prefix | Prefix of the file name for the head observations |
| file_head_suffix | Suffix of the file name for the head observations |
| state_type | Define variables included in the state vector |
| rms_obs_head | Observation error value used for the head |
| head_fixed_rmse | Whether to use a fixed value or the error values provided from the head observation file[*] |
| ResultPath | Path to the DA output file(s) |
| assim_o_sat | Whether to assimilation the soil water saturation observations[*] |
| path_obs_sat | Path to the file storing the soil water saturation observations |
| file_sat_prefix | Prefix of the file name for the soil water saturation observations |
| file_sat_suffix | Suffix of the file name for the soil water saturation observations |
| rms_obs_sat | Standard deviation value used for the soil water saturation observations |
| sat_fixed_rmse | Whether to use a fixed value or the error values provided from the soil water saturation observation file[*] |
| damp_k | Damping factor for hydraulic conductivity |
| Sr | Maximum saturation degree |

[*] 'True' if yes, and 'false' if no.



## Code and data availability

The current version of HGS-PDAF is available from GitHub https://github.com/qiqi1023t/HGS-PDAF_v1.0_GMD under the GNU General Public License v3.0. The exact version of the model used to produce the results used in this paper is
archived on Zenodo (https://zenodo.org/doi/10.5281/zenodo.10000886) (Tang et al., 2023), as are input data and scripts to run the model and produce the plots for all the simulations presented in this paper.

## Author contributions

The project was conceptualised by QT, PB and OS. Data curation was performed by QT and OS. Code development of the HGS-PDAF was carried out by QT, OS and WK. The HGS model used as an illustrative example was built by HD.
Numerical experiments based on this HGS model and HGS-PDAF code on the supercomputer were performed by QT. Formal analysis and figure visualisation were performed by QT and HD. The original draft was written by QT, HD and OS. The manuscript was revised and edited by OS, HD, QT, LN, WK and PB. Funding acquisition was carried out by PB and OS. Supervision was carried out by PB.

## Competing interests

One of the (co-)authors is a member of the editorial board of Geoscientific Model Development.

## Acknowledgement

Q. Tang gratefully acknowledges the funding from the European Union's Horizon 2020 project WATERAGRI (https://wateragri.eu, grant No. 858375). The authors acknowledge PRACE for awarding access to the Fenix Infrastructure
resources at Forschungszentrum Jülich, which are partially funded from the European Union's Horizon 2020 research and innovation programme through the ICEI project under the grant agreement No. ICEI-PRACE-2023-0004.

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
