# Peer review of "HGS-PDAF (version 1.0): A modular data assimilation framework for an integrated surface and subsurface hydrological model"

_Geoscientific Model Development, 2023_

## Referee Comment (RC3)

Review for manuscript: *HGS-PDAF (version 1.0): A modular data assimilation framework for an integrated surface and subsurface hydrological model*

**Summary:**
This manuscript integrates the Parallel Data Assimilation Framework (PDAF), an open-source data assimilation software, with the HydroGeoSphere (HGS) hydrological model. The integration involves separate and alternating executions of HGS and PDAF, enabling information from one model to inform the other. Similar applications involve combining EnKF with HGS and combining PDAF with ParFlow. PDAF encompasses two fundamental classes of DA methods: Ensemble Kalman Filter (EnKF)-based, offering distributions for estimators, and variational-based, providing point estimators. This study specifically demonstrates the model binding using EnKF-based PDAF, validated through an application in a quasi-hypothetical numerical river-aquifer model. The model's performance on state variables, hydraulic head and soil moisture, is assessed using their ensemble mean. The model parameter, hydraulic conductivity (K), is constrained by the expected prior distribution, aligning with the method's anticipated behavior.

**Comments:**
Line 21-23: The assertion of operational real-time management may be perceived as over-promising. It heavily depends on the infrastructure of data warehousing and model pipelines.

EnKF related:

- Line 169: Clarify the term "state vector with model parameters." Is Xp representing model parameters sampled at a given realization from its latent distribution?
- Equation 2: Specify whether the forward transient process is noise-free. While understanding that the noise term may be controlled by parameters in Xp, consider presenting EnKF in the standard state space model format, clearly defining states, parameters, and distributions.
- Equation 3: Define the observation model here to maintain a consistent format with Equation 2, rather than introducing it directly from Equation 4.
- Line 200-206: If parameters and states are well-defined, refer to them in this section. Consider adding this information to the suggested flowchart to visually represent the requirements.
- Line 279: I am curious whether the covariance matrix encounters degeneracy problems after many time steps.

Flowchart related:

- Lines 134-152: Clarity in this paragraph could be enhanced with the inclusion of a flowchart, similar to Figure 1.
- Consider improving the flowchart quality in Figures 1 to 4 by incorporating consistent boxes and colors to distinguish observation, model run, configuration, and output steps. Providing a flowchart illustrating the connections between different modules can offer a more comprehensive overview.

Figure 8: Strengthen your claim by plotting the standard error for ensemble mean, demonstrating statistical significance in error reduction to bolster your argument.

---

## Author Response (AR1)

Dear Topic Editor,

We would like to express our sincere thanks for your time and expertise in managing the review process for our manuscript. We would also like to thank the reviewers for taking the time and effort to provide detailed comments and suggestions. We have revised the paper according to the reviewers' comments, and our response to the comments is shown in blue and italics (line numbers refer to the line number in the new, clean version of the manuscript).

Sincerely,

Qi Tang

**Reviewer 1**

This paper by Tang et al. describes a data assimilation system for integrated surface-subsurface hydrologic models, that is capable of assimilating multivariate observations and performing dual state-parameter estimates. The data assimilation system is demonstrated with an ensemble Kalman filter, at a small domain, in a synthetic data experiment, with hydraulic heads and volumetric soil moisture assimilated. The paper is very well written, with the system clearly described and well demonstrated. However, I do have some concerns about the paper:

1. The assimilation interval seems to be one day (it is not very clear from the manuscript: the authors mentioned obtaining daily synthetic observations but did not explicitly mention the assimilation interval), and the errors are evaluated every day. Given the frequent assimilation of observations, it is difficult to evaluate whether the updates of states and parameters truly improved prediction skills. For example, if the observations are assimilated every three days, can the DA runs outperform the open loop runs?

*The assimilation interval in this illustrative example is one day. We have added a sentence in the manuscript to clarify this:*

*Line 460: "The assimilation interval is one day."*

*Since the focus of this paper is to show the structure of the HGS-PDAF framework, the synthetic experiment shown in Section 4 is purely an illustrative exercise that demonstrates how DA can be achieved via HGS-PDAF, nothing can be generalised from such a synthetic model. We agree that the assimilation frequency can have an influence on the assimilation performance, but since this is entirely illustrative, conducting an analysis of the effect of the assimilation frequency on updating is beyond the scope of this paper.*

2. Many hydrologic models are designed to improve flood/drought predictions, which means that stream discharge is the most important prediction. I feel the manuscript could be strengthened by a demonstration of either assimilating discharge observations, or improving predictions of discharge.

*As outlined in the reply to the previous comment, the illustrative example is essentially a purely synthetic case tailored towards demonstrating the modular capabilities of HGS-PDAF, not a real world or DA experiment analysing the effects of DA of different observation types on different hydrological*

*predictions. Assimilation/updating of other variables/types of observations such as stream discharge is of course possible with HGS-PDAF and was mentioned in the original manuscript in the conclusions section. As an illustrative case in the paper, we selected two types of observations that are important observations for the hydrogeological modeling and which allow demonstrating how DA can be achieved for HGS with HGS-PDAF. Extending this to more variables/observations is beyond the scope of this methods-oriented paper.*

3. It is not clear to me how hydraulic heads and soil water contents are updated separately. Hydraulic heads and soil water contents are connected by water retention curves. If they are updated simultaneously by EnKF, what is being used as initial conditions for the next prediction cycle? This is not explained in the manuscript.

*We thank the reviewer for pointing out to us that this was not stated clearly enough in the manuscript. When hydraulic head and soil water content (in terms of saturation as saturation is the directly used variable in HGS) are updated, they are both combined in the state vector and updated simultaneously using the covariance matrix. In the example shown in the paper, when these two variables are updated together, the initial condition for the next prediction cycle was only based on hydraulic head. This is now explained in the manuscript:*

*Lines 460-461: "When hydraulic heads and soil water saturation are updated together, the initial condition for the next prediction cycle is only hydraulic head."*

*We would like to state that the functional relationship between saturation and hydraulic head suggested by the reviewer is only applicable if unsaturated conditions are present. If the groundwater level rises, the head can still change yet the degree of saturation will be at 100%. As we are jointly simulating saturated/unsaturated conditions it is important to consider both saturation and head. Note also that the functional relationships are often associated with large uncertainties and processes such a hysteresis, which is not considered in our models. The consideration of these two variables is therefore not necessarily redundant. Given that our case is a purely illustrative example to demonstrate the modularity of HGS-PDAF, it is therefore out of scope of the paper to analyse the effects of different DA strategies when assimilating both hydraulic heads and soil water saturation simultaneously.*

4. I am very curious about why assimilating soil moisture content does not seem to improve the estimates. Have the authors checked the spread of hydraulic heads and soil water saturation of the ensemble, and compared with the errors of hydraulic heads and soil water saturation observations? I feel that my last concern could be also related to this problem.

*In this specific example, we did not explicitly simulate evapotranspiration, and the thin unsaturated zone thus only exists when the groundwater level decreases. Therefore, assimilating the soil water saturation only affects the unsaturated zone while assimilating the heads will change the head in both the saturated and unsaturated zones. In this very specific example, therefore, water saturation is not as informative as hydraulic heads. However, again we would like to stress that the focus of this paper was to show the structure of the HGS-PDAF framework. By considering saturation observations we intended to show the possibility of assimilating multiple observation types simultaneously (i.e. multi-variate assimilation), in this case hydraulic heads and soil water saturation. The point of the example is not to investigate the suitability of DA strategies, or the effect (and many difficult choices to be made) by jointly updating of hydraulic heads and soil water saturation. Doing this would be out of the scope of this paper.*

Specific comments

1.  L83: "the coupling was neither modular nor user-friendly for…" "For" is redundant.

*Deleted as suggested by the reviewer.*

2.  L89: "PDAF makes it very easy to switch between different assimilation methods without the need for additional coding." Does the observation array needs to be re-coded if the assimilation method has changed?

*No, the observation array is independent of the assimilation method. Changing the assimilation method within the PDAF doesn't affect the observation array.*

3.  In the manuscript, hydraulic conductivity is chosen to be modified. Based on my past experience, the parameters controlling the water retention curves can be even more important. Have the authors considered this?

*We agree that the parameters controlling the water retention curves, such as alpha and n in the van Genuchten model, are important when considering variably saturated flow in the aquifer. In our case we have predefined the pressure-saturation relationship table and therefore the water retention curve doesn't change during the assimilation. These parameters are taken into account and can be flexibly added to the HGS-PDAF framework in future applications.*

4.  L129: "a dual dual-aquifer configuration."

*The first dual is deleted.*

5.  L159: "typical states that are considered for updating are hydraulic heads, surface water discharge, soil moisture, evapotranspiration or solute concentrations." Discharge and evapotranspiration are not states, but fluxes.

*Corrected as suggested by the reviewer.*

6.  Equation (2) "the observations are perturbed by a reasonably chosen representative observation error." This is interesting. I don't think the classic EnKF requires the perturbation of assimilated observations though.

*The classical EnKF requires perturbed observations. This was clarified by Burgers et al. (1998) but it was missing in Evensen (1994).*

7.  L181: The authors may need to explain the term filter divergence. People outside of the DA community may not know what it stands for.

*We have added one sentence to explain filter divergence:*

*Lines 187-190: "Filter divergence refers to the situation where the estimated state of the system becomes increasingly inaccurate or divergent from the true state over time. This divergence occurs when the filtering algorithm fails to effectively incorporate new observations or when the model's dynamics do not properly represent the underlying system."*

8. L431: "These observation time series were subsequently stochastically perturbed by a normally distributed error with a standard deviation of 5cm for hydraulic heads and 1% for soil water saturation." How were the errors determined?

*These observation errors are based on the prior knowledge and the tuning experiments, e.g. 5 % and 10 % have also been tested as the saturation error. As the example shown in the paper is based on a synthetic model setup and the observations are also generated synthetically, we use a relatively small observation error to better illustrate how HGS-PDAF works. We appreciate the reviewer's suggestion and this has now been clarified in the manuscript:*

*Lines 445-447: "The values of the observation errors are determined by our prior knowledge and tuning experiments. Different percentages such as 5% and 10% were tested and subsequently defined to provide a most illustrative use case."*

9. Figure 9: I must admit I got lost when looking at Figure 9. I am not sure what the x- and y-axes are. They look like spatial maps and I assume they are the spatial x and y directions but I am not sure.

*Yes, the original Figure 9 (which in the revised manuscript is now Figure 10) is a spatial map of the model domain. We have added the x- and y- axis legend as well as the flow direction for this figure. We hope it is now clear.*

**Reviewer 2**

In this manuscript, the authors present a coupling framework to integrate a data assimilation toolbox with the HydroGeoSphere (HGS) fully-coupled groundwater – surface water model. This is timely work as there is increasing interest in operationalizing structurally and physically complex models like HGS, and robust data assimilation methodology is required. The manuscript is suited for GMD, well written, and in general, well organized. I only have a few minor comments for the authors to consider.

L101: this bullet point needs clarified.

*This is now clarified:*

*Lines 99-101: "2) a modular tool to handle different types of observation data, which enables to assimilate one or multiple types of observations simultaneously, currently programmed for hydraulic heads, soil moisture and solute concentration measurements."*

L123: Replace 'Saint-Venant' with 'diffusion wave'. Could also mention one-dimensional open channel flow.

*We replaced this as suggested by the reviewer. In HGS, the surface water flow is represented as two-dimensional depth-averaged areal flow.*

L140: Comma not needed behind 'files'.

*Corrected as suggested by the reviewer.*

L165: multiple realizations of a numerical model.

*Corrected as suggested by the reviewer.*

L304: values for the nodes (I believe these are nodal properties referred to in this sentence).

*Corrected as suggested by the reviewer.*

L364: What is the clock speed for these CPUs? Were the individual HGS simulations also parallelized, if so, across how many cores?

*The clock speed per computing node is 2.25 GHz. No, the individual HGS simulation is not parallelized, i.e. each HGS model is run on 1 core. This is now clarified in the manuscript:*

*Lines 374-375: "The clock speed per computing node is 2.25 GHz. The individual HGS simulation is not parallelized, i.e. each HGS model is run on 1 core."*

L391: river bank filtration pumping wells,

*Corrected as suggested by the reviewer.*

L412: (tint)

*Removed as suggested by the reviewer.*

L414: could remove (i.e. with maximum pumping regime)

*Removed as suggested by the reviewer.*

L417: brackets around (i.e. K).

*Corrected as suggested by the reviewer.*

L423: producing a heterogeneous parameter field.

*Corrected as suggested by the reviewer.*

L430: would saturation at these points not be dependent on head, hence head and saturation at coincident points is redundant?

*The saturation and hydraulic head depend on each other in the unsaturated zone. When hydraulic head and soil water saturation are updated, they are both combined in the state vector and updated simultaneously using the covariance matrix. In the example shown in the paper, when these two variables are updated together, the initial condition for the next prediction cycle was only based on hydraulic head. This is now explained in the manuscript:*

*Lines 460-461: "When hydraulic heads and soil water saturation are updated together, the initial condition for the next prediction cycle is only hydraulic head."*

*We would like to state that the functional relationship between saturation and hydraulic head suggested by the reviewer is only applicable if unsaturated conditions are present. If the groundwater*

*level rises, the head can still change yet the degree of saturation will be at 100%. As we are jointly simulating saturated/unsaturated conditions it is important to consider both saturation and head. Note also that the functional relationships are often associated with large uncertainties and processes such a hysteresis, which is not considered in our models. The consideration of these two variables is therefore not necessarily redundant. Given that our case is a purely illustrative example to demonstrate the modularity of HGS-PDAF, it is therefore out of scope of the paper to analyse the effects of different DA strategies when assimilating both hydraulic heads and saturation simultaneously.*

L433: This perturbation is quite small in relation to variability in a natural system of similar scale, and in particular 1 % SD in moisture content is almost negligible. Could the authors comment on what would be considered reasonable values for a real-world scenario, and how run times might be affected?

*We agree that 1% SD is low for a real-test case study where the spatial representation of measured saturations may be influenced by local scale heterogeneities and preferential flow paths. However, the example shown in the paper is based on a synthetic model setup and the observations are also generated synthetically, we determine the observation error values based on the prior knowledge and the tuning experiments, e.g. 5 % and 10 % have also been tested as the saturation error. We use a relatively small observation error to better illustrate how HGS-PDAF works. We have added a few sentences to describe this in the manuscript:*

*Lines 445-447: "The values of the observation errors are determined by our prior knowledge and tuning experiments. Different percentages such as 5% and 10% were tested and subsequently defined to provide a most illustrative use case."*

*In a real-world scenario, such a measurement error may be higher to also account for measurement representation. Specific values for measurement errors are case specific but must always respect proper balance between goodness of fit and over-fitting to preserve the consistency of the updated states.*

*In terms of runtimes, HGS uses adaptive time steps for numerical iteration. Once DA is implemented, as described in the manuscript, the simulation will be interrupted per assimilation frequency and the model will always need to be restarted and initialised with an initial-small time step. This will certainly increase the overall run times.*

General comment:

- Could the authors comment in the manuscript on how perturbations in head and moisture content affected the numerical stability and time-step intervals for subsequent simulations? Is there a sweet spot for the amount of perturbation so that both data assimilation and model run times can be optimized? It is my understanding that if updates to the model state induce shocks or instabilities into the initial condition then simulation run times can appreciably slow down.

*For this synthetic model we have tested different saturation error values such as 1%, 5% and 10% to monitor the model stability against the assimilation performance and to define an optimal error to to achieve a most illustrative use case. The total simulation run times are similar for the three cases with different observation errors, while the best results are obtained with the smallest observation error, i.e. 1%. As this example is based on a synthetic model setup, and the observations are also generated synthetically, such a small observation error doesn't induce shocks or instabilities in the initial condition and therefore doesn't significantly increase the simulation runtimes, so the sweet spot depends only on the assimilation performance. However, since the focus of this paper is to show the*

*structure of the developed HGS-PDAF framework, and this synthetic experiment is purely an illustrative exercise to show how DA can be achieved via HGS-PDAF, nothing can be generalised from such a synthetic model. We agree that in a real case, the deviation between the model simulation and the real observation can be large, and updating the model state with a small observation error can affect the numerical stability, thus increasing the time step intervals and the total simulation run times. However, as this is purely illustrative, to carry out an analysis of perturbations and time steps on data assimilation performance is beyond the scope of this paper.*

- EnKF has been used now for a number of HGS DA applications. However, as the authors note, the PDAF toolbox supports many other DA approaches. Could the authors add a table to the manuscript that lists the other DA approaches, previous application of these approaches towards hydrologic modeling, and general guidelines for users of the HGS-PDAF framework to select the most suitable approach for their application? Or perhaps list the strengths and weaknesses of the different approaches WRT fully coupled groundwater – surface water modeling?

*We have added a table (Appendix 2) as suggested by the reviewer. It shows the DA approaches supported by PDAF, field of application and examples of reference. DA approaches are application dependent, and the classical EnKF should be fine when the number of observations is rather low, as in our illustrative example, and is therefore widely used in hydrological simulation. If the observation number is high, we can also consider different types of ensemble transform Kalman filters, such as ETKF (Bishop et al., 2001) and ESTKF (Nerger et al., 2012). In particular, if the number of observations is large, localisation (Nerger et al., 2006) should also be considered.*

*Lines 201-202: "The available DA approaches and their application fields as well as several example references are listed in Appendix 1."*

*"Appendix 1: Data assimilation approaches in PDAF and their known application fields*

| Data assimilation approaches | | | Fields of application | Examples in hydrogeology (if not applicable, we give references in other fields and marked with *) |
|---|---|---|---|---|
| *Ensemble based* | *Global* | *EnKF* | *Meteorology, oceanography, hydrology, hydrogeology, land surface* | *Tang et al. (2017); Tang et al. (2018)* |
| | | *ETKF* | *Meteorology, oceanography, hydrology, hydrogeology, land surface* | *Rasmussen et al. (2016); Zhang et al. (2016)* |
| | | *SEIK* | *Meteorology, oceanography, hydrology, hydrogeology* | *Schumacher (2016)* |
| | | *ESTKF* | *Meteorology, oceanography, hydrology, hydrogeology* | *Li et al. (2023b)* |
| | | *NETF* | *Meteorology, oceanography* | *Nerger (2022); Tödter et al. (2016)** |
| | | *PF* | *Meteorology, oceanography, hydrology, hydrogeology, land surface* | *Abbaszadeh et al. (2018); Berg et al. (2019)* |

| | | SEEK | Meteorology, oceanography | Brasseur and Verron (2006); Butenschön and Zavatarelli (2012)* |
|---|---|---|---|---|
| | Local | LEnKF | Meteorology, oceanography, hydrology, hydrogeology, land surface | Hung et al. (2022); Li et al. (2023a) |
| | | LETKF | Meteorology, oceanography, hydrology, hydrogeology, land surface | Sawada (2020) |
| | | LSEIK | Meteorology, oceanography | Liang et al. (2017); Liu and Fu (2018)* |
| | | LESTKF | Meteorology, oceanography | Zheng et al. (2020)* |
| | | LNETF | Meteorology, oceanography | Feng et al. (2020)* |
| | | LKNETF | Meteorology, oceanography | Shao and Nerger (2024)* |
| Variational | | 3DVAR | Meteorology, oceanography, hydrology | Cummings and Smedstad (2013); Li et al. (2008)* |

*"*

**Reviewer 3**

 **Summary:**
This manuscript integrates the Parallel Data Assimilation Framework (PDAF), an open-source data assimilation software, with the HydroGeoSphere (HGS) hydrological model. The integration involves separate and alternating executions of HGS and PDAF, enabling information from one model to inform the other. Similar applications involve combining EnKF with HGS and combining PDAF with ParFlow. PDAF encompasses two fundamental classes of DA methods: Ensemble Kalman Filter (EnKF)-based, offering distributions for estimators, and variational-based, providing point estimators. This study specifically demonstrates the model binding using EnKF-based PDAF, validated through an application in a quasi-hypothetical numerical river-aquifer model. The model's performance on state variables, hydraulic head and soil moisture, is assessed using their ensemble mean. The model parameter, hydraulic conductivity (K), is constrained by the expected prior distribution, aligning with the method's anticipated behavior.

**Comments:**
Line 21-23: The assertion of operational real-time management may be perceived as over-promising. It heavily depends on the infrastructure of data warehousing and model pipelines.

*We couldn't agree more with the reviewer that operational real-time management requires much more than just a model and a DA platform. It also requires sensors, secure and robust data transmission and storage, other infrastructure and pipelines. However, what we assert with our statement is that with the integrated model and this modular DA framework, we have essentially developed the hydrologically and DA wise robust toolbox for developing the basic model for operational management of coupled surface water-groundwater resources. We have adjusted the statement accordingly:*

*Lines 21-23: "With the integrated model and this modular DA framework, we have essentially developed the hydrologically and DA wise robust toolbox for developing the basic model for operational management of coupled surface water-groundwater resources."*

EnKF related:
- Line 169: Clarify the term "state vector with model parameters." Is Xp representing model parameters sampled at a given realization from its latent distribution?

*Yes, the term Xp represents the model parameters from a given distribution. To clarify the state vector, we reformulated the equation to describe the state vector:*

*Lines 166-175: "In mathematical terms, consider that a state vector $\boldsymbol{X}$ can be written as Eq. (1):*

$$X_i = (X_s) \tag{1}$$

*where $\boldsymbol{X_s}$ is the state vector with model state variables. When parameters are updated together with the state variables, the augmented state vector can be written as*

$$X_i = \begin{pmatrix} X_S \\ X_p \end{pmatrix}_i \tag{2}$$

*where $\mathbf{X}_p$ is the state vector with model parameters."*

- Equation 2: Specify whether the forward transient process is noise-free. While understanding that the noise term may be controlled by parameters in Xp, consider presenting EnKF in the standard state space model format, clearly defining states, parameters, and distributions.

*Yes, it's noise free. We have reformulated the equation to describe the state vector. Please see our previous point.*

- Equation 3: Define the observation model here to maintain a consistent format with Equation 2, rather than introducing it directly from Equation 4.

*We agree that this is not the original version of the observation model which maps the observations to the model state but only to described how the observations are perturbed by the observation errors in EnKF. This has been explained in Burgers et al. (1998). In order to maintain the consistency of such a modified version of EnKF, we leave this formula here.*

- Line 200-206: If parameters and states are well-defined, refer to them in this section. Consider adding this information to the suggested flowchart to visually represent the requirements.

*We added this information in the manuscript as suggested by the reviewer:*

*Lines 212-215: "whether the model parameters are included in the state vector for updating along with the state variables. If yes, and if the parameters to be included is the hydraulic conductivity (K),"*

*As this is also related to the Flowchart of the initialisation of data assimilation, we also updated the corresponding description text:*

*Line 317-319: "Notice that we may need transferring the original values of the model state or parameters, e.g. for K, the log-transformed K is considered in the state vector rather than the K itself used in the HGS model to ensure that K is always positive during the assimilation process;"*

*The flowchart itself is not changed as this is part of the definition of the state vector.*

• Line 279: I am curious whether the covariance matrix encounters degeneracy problems after many time steps.

*In Figure 9 the two realisations of K are different which indicates the covariance matrix is not too small, which in turn means that until the end of the simulation period, there is no covariance matrix degeneracy problem.*

Flowchart related:
• Lines 134-152: Clarity in this paragraph could be enhanced with the inclusion of a flowchart, similar to Figure 1.

*We have added a flowchart (new Figure 1) to clarify the workflow of HydroGeoSphere.*

• Consider improving the flowchart quality in Figures 1 to 4 by incorporating consistent boxes and colors to distinguish observation, model run, configuration, and output steps. Providing a flowchart illustrating the connections between different modules can offer a more comprehensive overview.

*In Figure 1-5, the green blocks are the HGS model related parts, the yellow blocks are the model bindings, the blue block is the PDAF software, and the orange blocks are the observation related parts. Figure 3 shows the connection between the different modules/subroutines. The parameter modules are not shown in this figure as they are not the process module and are predefined and used by the initialisation subroutines. This is already described in the manuscript in section 3.3.2.*

Figure 8: Strengthen your claim by plotting the standard error for ensemble mean, demonstrating statistical significance in error reduction to bolster your argument.

*Currently we only store the ensemble mean for the variables in the state vector and the standard deviation if not saved. Thus, plotting the standard error is currently not possible.*

Reference:

Abbaszadeh, P., Moradkhani, H., and Yan, H., 2018, Enhancing hydrologic data assimilation by evolutionary Particle Filter and Markov Chain Monte Carlo: Advances in Water Resources, v. 111, p. 192-204.

Berg, D., Bauser, H. H., and Roth, K., 2019, Covariance resampling for particle filter – state and parameter estimation for soil hydrology: Hydrol. Earth Syst. Sci., v. 23, no. 2, p. 1163-1178.

Bishop, C. H., Etherton, B. J., and Majumdar, S. J., 2001, Adaptive Sampling with the Ensemble Transform Kalman Filter. Part I: Theoretical Aspects: Monthly Weather Review, v. 129, no. 3, p. 420-436.

Brasseur, P., and Verron, J., 2006, The SEEK filter method for data assimilation in oceanography: a synthesis: Ocean Dynamics, v. 56, no. 5, p. 650-661.

Burgers, G., Jan van Leeuwen, P., and Evensen, G., 1998, Analysis Scheme in the Ensemble Kalman Filter: Monthly Weather Review, v. 126, no. 6, p. 1719-1724.

Butenschön, M., and Zavatarelli, M., 2012, A comparison of different versions of the SEEK Filter for assimilation of biogeochemical data in numerical models of marine ecosystem dynamics: Ocean Modelling, v. 54-55, p. 37-54.

Cummings, J. A., and Smedstad, O. M., 2013, Variational Data Assimilation for the Global Ocean, *in* Park, S. K., and Xu, L., eds., Data Assimilation for Atmospheric, Oceanic and Hydrologic Applications (Vol. II): Berlin, Heidelberg, Springer Berlin Heidelberg, p. 303-343.

Evensen, G., 1994, Sequential data assimilation with a nonlinear quasi-geostrophic model using Monte Carlo methods to forecast error statistics: Journal of Geophysical Research: Oceans, v. 99, no. C5, p. 10143-10162.

Feng, J., Wang, X., and Poterjoy, J., 2020, A Comparison of Two Local Moment-Matching Nonlinear Filters: Local Particle Filter (LPF) and Local Nonlinear Ensemble Transform Filter (LNETF): Monthly Weather Review, v. 148, no. 11, p. 4377-4395.

Hung, C. P., Schalge, B., Baroni, G., Vereecken, H., and Hendricks Franssen, H.-J., 2022, Assimilation of Groundwater Level and Soil Moisture Data in an Integrated Land Surface-Subsurface Model for Southwestern Germany: Water Resources Research, v. 58, no. 6, p. e2021WR031549.

Li, F., Kurtz, W., Hung, C. P., Vereecken, H., and Hendricks Franssen, H.-J., 2023a, Water table depth assimilation in integrated terrestrial system models at the larger catchment scale: Frontiers in Water, v. 5.

Li, Y., Cong, Z., and Yang, D., 2023b, Remotely Sensed Soil Moisture Assimilation in the Distributed Hydrological Model Based on the Error Subspace Transform Kalman Filter: Remote Sensing, v. 15, no. 7, p. 1852.

Li, Z., Chao, Y., McWilliams, J. C., and Ide, K., 2008, A Three-Dimensional Variational Data Assimilation Scheme for the Regional Ocean Modeling System: Journal of Atmospheric and Oceanic Technology, v. 25, no. 11, p. 2074-2090.

Liang, X., Yang, Q., Nerger, L., Losa, S. N., Zhao, B., Zheng, F., Zhang, L., and Wu, L., 2017, Assimilating Copernicus SST Data into a Pan-Arctic Ice–Ocean Coupled Model with a Local SEIK Filter: Journal of Atmospheric and Oceanic Technology, v. 34, no. 9, p. 1985-1999.

Liu, Y., and Fu, W., 2018, Assimilating high-resolution sea surface temperature data improves the ocean forecast potential in the Baltic Sea: Ocean Sci., v. 14, no. 3, p. 525-541.

Nerger, L., 2022, Data assimilation for nonlinear systems with a hybrid nonlinear Kalman ensemble transform filter: Quarterly Journal of the Royal Meteorological Society, v. 148, no. 743, p. 620-640.

Nerger, L., Danilov, S., Hiller, W., and Schröter, J., 2006, Using sea-level data to constrain a finite-element primitive-equation ocean model with a local SEIK filter: Ocean Dynamics, v. 56, no. 5, p. 634-649.

Nerger, L., Janjić, T., Schröter, J., and Hiller, W., 2012, A Unification of Ensemble Square Root Kalman Filters: Monthly Weather Review, v. 140, no. 7, p. 2335-2345.

Rasmussen, J., Madsen, H., Jensen, K. H., and Refsgaard, J. C., 2016, Data assimilation in integrated hydrological modelling in the presence of observation bias: Hydrol. Earth Syst. Sci., v. 20, no. 5, p. 2103-2118.

Sawada, Y., 2020, Do surface lateral flows matter for data assimilation of soil moisture observations into hyperresolution land models?: Hydrol. Earth Syst. Sci., v. 24, no. 8, p. 3881-3898.

Schumacher, M., 2016, Methods for assimilating remotely-sensed water storage changes into hydrological models: Universitäts-und Landesbibliothek Bonn.

Shao, C., and Nerger, L., 2024, The Impact of Profiles Data Assimilation on an Ideal Tropical Cyclone Case: Remote Sensing, v. 16, no. 2, p. 430.

Tang, Q., Kurtz, W., Schilling, O. S., Brunner, P., Vereecken, H., and Hendricks Franssen, H. J., 2017, The influence of riverbed heterogeneity patterns on river-aquifer exchange fluxes under different connection regimes: Journal of Hydrology, v. 554, p. 383-396.

Tang, Q., Schilling, O. S., Kurtz, W., Brunner, P., Vereecken, H., and Hendricks Franssen, H.-J., 2018, Simulating Flood-Induced Riverbed Transience Using Unmanned Aerial Vehicles, Physically Based Hydrological Modeling, and the Ensemble Kalman Filter: Water Resources Research, v. 54, no. 11, p. 9342-9363.

Tödter, J., Kirchgessner, P., Nerger, L., and Ahrens, B., 2016, Assessment of a Nonlinear Ensemble Transform Filter for High-Dimensional Data Assimilation: Monthly Weather Review, v. 144, no. 1, p. 409-427.

Zhang, D., Madsen, H., Ridler, M. E., Kidmose, J., Jensen, K. H., and Refsgaard, J. C., 2016, Multivariate hydrological data assimilation of soil moisture and groundwater head: Hydrol. Earth Syst. Sci., v. 20, no. 10, p. 4341-4357.

Zheng, Y., Albergel, C., Munier, S., Bonan, B., and Calvet, J. C., 2020, An offline framework for high-dimensional ensemble Kalman filters to reduce the time to solution: Geosci. Model Dev., v. 13, no. 8, p. 3607-3625.